# Long-Term PDE-5A Inhibition Improves Myofilament Function in Left and Right Ventricular Cardiomyocytes through Partially Different Mechanisms in Diabetic Rat Hearts

**DOI:** 10.3390/antiox10111776

**Published:** 2021-11-06

**Authors:** Beáta Bódi, Árpád Kovács, Hajnalka Gulyás, Lilla Mártha, Attila Tóth, Csaba Mátyás, Bálint András Barta, Attila Oláh, Béla Merkely, Tamás Radovits, Zoltán Papp

**Affiliations:** 1Division of Clinical Physiology, Department of Cardiology, Faculty of Medicine, University of Debrecen, 4032 Debrecen, Hungary; bodibea0509@gmail.com (B.B.); kovacs.arpad@med.unideb.hu (Á.K.); gulyas.hajnalka@med.unideb.hu (H.G.); lilla.martha@med.unideb.hu (L.M.); atitoth@med.unideb.hu (A.T.); 2Doctoral School of Pharmaceutical Sciences, University of Debrecen, 4032 Debrecen, Hungary; 3HAS-UD Vascular Biology and Myocardial Pathophysiology Research Group, Hungarian Academy of Sciences, 4032 Debrecen, Hungary; 4Heart and Vascular Center, Semmelweis University, 1122 Budapest, Hungary; csmatyas@yahoo.com (C.M.); barta.balint@gmail.com (B.A.B.); o.attilio@gmail.com (A.O.); merkely.bela@gmail.com (B.M.); radovitstamas@yahoo.com (T.R.)

**Keywords:** right ventricle, HFpEF, diabetic cardiomyopathy, cardiomyocyte passive tension, Ca^2+^-sensitivity of force production, myofilament protein phosphorylation, vardenafil, phosphodiesterase-5A

## Abstract

Heart failure with preserved ejection fraction (HFpEF) and right ventricular (RV) dysfunction are frequent complications of diabetic cardiomyopathy. Here we aimed to characterize RV and left ventricular (LV) remodeling and its prevention by vardenafil (a long-acting phosphodiesterase-5A (PDE-5A) inhibitor) administration in a diabetic HFpEF model. Zucker Diabetic Fatty (ZDF) and control, ZDF Lean (Lean) male rats received 10 mg/kg vardenafil (ZDF + Vard; Lean + Vard) per os, on a daily basis for a period of 25 weeks. In vitro force measurements, biochemical and histochemical assays were employed to assess cardiomyocyte function and signaling. Vardenafil treatment increased cyclic guanosine monophosphate (cGMP) levels and decreased 3-nitrotyrosine (3-NT) levels in the left and right ventricles of ZDF animals, but not in Lean animals. Cardiomyocyte passive tension (F_passive_) was higher in LV and RV cardiomyocytes of ZDF rats than in those receiving preventive vardenafil treatment. Levels of overall titin phosphorylation did not differ in the four experimental groups. Maximal Ca^2+^-activated force (F_max_) of LV and RV cardiomyocytes were preserved in ZDF animals. Ca^2+^-sensitivity of isometric force production (pCa_50_) was significantly higher in LV (but not in RV) cardiomyocytes of ZDF rats than in their counterparts in the Lean or Lean + Vard groups. In accordance, the phosphorylation levels of cardiac troponin I (cTnI) and myosin binding protein-C (cMyBP-C) were lower in LV (but not in RV) cardiomyocytes of ZDF animals than in their counterparts of the Lean or Lean + Vard groups. Vardenafil treatment normalized pCa_50_ values in LV cardiomyocytes, and it decreased pCa_50_ below control levels in RV cardiomyocytes in the ZDF + Vard group. Our data illustrate partially overlapping myofilament protein alterations for LV and RV cardiomyocytes in diabetic rat hearts upon long-term PDE-5A inhibition. While uniform patterns in cGMP, 3-NT and F_passive_ levels predict identical effects of vardenafil therapy for the diastolic function in both ventricles, the uneven cTnI, cMyBP-C phosphorylation levels and pCa_50_ values implicate different responses for the systolic function.

## 1. Introduction

The prevalence of metabolic disorders such as obesity and type 2 diabetes mellitus (T2DM) and the burden of heart failure (HF) increase worldwide [1]. T2DM is complicated by diabetic cardiomyopathy presenting with left ventricular (LV) diastolic dysfunction: i.e., increased LV stiffness with elevated end-diastolic pressure and impaired LV filling [2]. Of note, LV diastolic dysfunction is also a hallmark of HF with preserved LV ejection fraction (HFpEF), a type of HF with poor therapeutic responses for conventional HF medications [3].

Worsening right ventricular (RV) performance is a common, and poor prognostic feature of HFpEF with a hypothetical RV cardiomyocyte dysfunction [4,5]. Moreover, metabolic syndrome may also present with biventricular dysfunction and HFpEF. Interestingly, diabetic and hypertensive patients first develop signs of subclinical RV diastolic dysfunction and thereafter systolic impairment [6,7,8]. As a matter of fact, the prognostic significance of declining RV function appears to be higher than that of LV performance in HFpEF patients [9]. Moreover, RV dilatation and dysfunction are more severe, and exercise capacity is worse in obese HFpEF patients than those in non-obese HFpEF patients, hence suggesting an obesity-related HFpEF subtype [10].

Nitro-oxidative stress is thought to downregulate the nitric oxide (NO)/soluble guanylyl cyclase (sGC)/cyclic guanosine monophosphate (cGMP)/protein kinase G (PKG) signaling pathway in metabolic models of HFpEF [11,12]. Oxidative stress also increases the expression of phosphodiesterase (PDE) 5, a negative regulator of the PKG pathway driving the breakdown of cGMP in the failing heart [13]. Therefore, pharmacological inhibition of PDE-5 has been considered as a promising therapeutic target in HFpEF [14]. The application of PDE-5A inhibitor sildenafil, however, was not effective either for the left [15] or the right heart [16] in the RELAX trial. In contrast, both cardiomyocyte protein alterations and LV diastolic dysfunction could be prevented by a chronic PDE-5A inhibitory therapy in animal models of HFpEF [17,18]. Nevertheless, the effects of preventive PDE-5A inhibitory therapy on RV cardiomyocyte function and signaling remain obscure.

Here we aimed to compare the functions and compositions of LV and RV cardiomyocytes in a diabetic rat model of HFpEF. We found mechanical and molecular differences between RV and LV cardiomyocytes, as well as distinct responses for long-term PDE-5A inhibitory therapy.

## 2. Materials and Methods

### 2.1. Animal Model

All animal care and experimental procedures involved in this work conformed strictly to the EU Directive 2010/63/EU of the European Parliament and the Guide for the Care and Use of Laboratory Animals used by the US National Institutes of Health (NIH Publication No. 85–23, revised 1996). The experimental protocol was approved by the institutional ethics committee of the Semmelweis University (permission number: PE/EA/418-4/2020).

Male 7-week-old Zucker diabetic fatty (ZDF) rats were studied. ZDF diabetic (fa/fa) and control Lean (+/?) rats (Charles River, Sulzfeld, Germany) were randomized into four groups: vehicle-treated controls (Lean; *n* = 8), vardenafil-treated controls (Lean + Vard; *n* = 7), vehicle-treated diabetic (ZDF; *n* = 7), and vardenafil-treated diabetic rats (ZDF + Vard; *n* = 8) [18]. Animals were fed with Purina #5008 diet (Charles River) and drank tap water *ad libitum*. Daily *per os* drug treatment [10 mg/kg body weight (BW) vardenafil dissolved in 0.01 M/L citrate buffer] or vehicle (0.01 M/L citrate buffer) administration via drinking water was initiated at the age of 7 weeks and continued until the end of the experimental period. Functional and biochemical measurements were performed. Hearts were excised, the left (LV) and right ventricle (RV) were dissected in cold saline solution on ice, placed in sterile cryovials, and immediately shock-frozen in liquid nitrogen at the age of 32 weeks. LV and RV tissue samples were thereafter stored at −80 °C.

### 2.2. Mechanical Measurements in Permeabilized Myocyte-Sized Preparations

Isometric force measurements were performed as previously described [19]. Deep-frozen LV and RV samples from all the four experimental groups were mechanically disrupted in isolating solution (ISO; 1 mM MgCl_2_, 100 mM KCl, 2 mM EGTA, 4 mM ATP, 10 mM imidazole; pH 7.0, 0.5 mM phenylmethylsulfonyl fluoride, 40 μM leupeptin and 10 μM E-64, all from Sigma-Aldrich, St. Louis, MO, USA) using a tissue homogenizer at 4 °C. Mechanically isolated cardiomyocytes were then subjected to incubation in ISO completed with 0.5% (*v*/*v*) Triton X-100 detergent (Sigma–Aldrich, St. Louis, MO, USA) for 5 min, which eliminated all membrane structures. Isolated and permeabilized single cardiomyocytes were attached with silicone adhesive (DAP 100% all-purpose silicone sealant; Baltimore, MD, USA) between two stainless insect needles, linked to a very sensitive force transducer (SensoNor, Horten, Norway) and an electromagnetic motor (Aurora Scientific Inc., Aurora, ON, Canada) in ISO at 15 °C. After adjustment of the sarcomere length to 2.3 μm, force measurements were performed by transferring the cells from relaxing solution (10 mM N,N-Bis (2-hydroxyethyl)-2-aminoethanesulfonic acid, 37.11 mM KCl, 6.41 mM MgCl_2_, 7 mM EGTA, 6.94 mM ATP, 15 mM creatine phosphate; pH 7.2 all from Sigma-Aldrich, St. Louis, MO, USA) to activating solution (same content as relaxing solution, supplemented with Ca^2+^-EGTA) in the presence of protease inhibitors: 0.5 mM phenylmethylsulfonyl fluoride, 40 μM leupeptin and 10 μM E-64. Ca^2+^ concentrations were expressed in pCa units calculated as follows: -lg[Ca^2+^]. The pCa of the activating and relaxing solutions was 4.75 and 9.0, respectively. Solutions with intermediate [Ca^2+^] were prepared by mixing relaxing and activating solutions (pCa 5.4–7.0). Ca^2+^-activated force (F_active_) values at intermediate [Ca^2+^] were normalized to maximal Ca^2+^-activated force (F_max_; at pCa 4.75) and then fitted to a modified Hill-equation (Origin 6.0, Microcal Software, Northampton, MA, USA), generating a sigmoid curve to define the Ca^2+^-sensitivity of the contractile system (pCa_50_). Ca^2+^-independent passive tension (F_passive_) of isolated cardiomyocytes was measured by shortening the cells to 80% of their initial lengths for 8 s in relaxing solution after a quick release–restretch maneuver (30 ms). Original forces of every individual cell were normalized to cardiomyocyte cross sectional-area, calculated by the width and height of the corresponding cardiomyocyte. Force values are expressed in kN/m^2^.

### 2.3. SDS-PAGE and Western Immunoblot Analysis

Frozen LV and RV samples were thawed and permeabilized as described above. Myocardial samples were thereafter homogenized in sample buffer (containing 8 M urea, 2 M thiourea, 3% (*w*/*v*) sodium dodecyl sulfate (SDS), 75 mM DTT, 50 mM Tris-HCl, pH 6.8, 10% (*v*/*v*) glycerol, bromophenol blue, 40 µM leupeptin and 10 µM E-64) for 45 min. Protein amounts of supernatants were determined after centrifugation (16,000× *g* for 5 min at 24 °C), using a dot-blot technique with bovine serum albumin (BSA) standard and the concentration of samples was adjusted to a final concentration of 2 mg/mL. Agarose-strengthened 2% SDS-polyacrilamide gels were used to separate N2B titin. Total phosphorylation status of titin protein was assessed by Pro-Q^®^ Diamond phosphoprotein staining (Invitrogen, Molecular Probes, Eugene, OR, USA) according to the manufacturer’s protocol, while total amount of protein was stained by Coomassie blue (Reanal, Budapest, Hungary).

Western immunoblot analyses were applied to assess site-specific phosphorylation levels of myofilament proteins. Separation of cardiac troponin I (cTnI) and cardiac myosin binding protein-C (cMyBP-C) was carried out in 4% and 12% polyacrilamide gels, respectively. After polyacrilamide gel electrophoresis and protein blotting procedure, membranes were blocked with 2% BSA diluted in PBS containing 0.1% (*v*/*v*) Tween 20 (PBST) for 30 min. Next, cTnI phosphorylation-sensitive antibodies were used to determine the levels of protein phosphorylation at protein kinase A (PKA)- and protein kinase C (PKC)-specific cTnI (Ser-23/24 (1:1000), Ser-43 (1:500) and Thr-143 (1:500), Abcam, Cambridge, UK) and PKA-specific cMyBP-C Ser-282 (1:500, Enzo Life Science, Farmingdale, NY, USA) phosphorylation sites. The signal was detected with a peroxidase-conjugated anti-rabbit IgG secondary antibody (1:300, Sigma-Aldrich, St. Louis, MO, USA) on nitrocellulose membranes. Total amount of proteins was visualized with super sensitive membrane stain (UD-GenoMed, Debrecen, Hungary). Signals were detected with an MF-ChemiBIS 3.2 gel documentation system (DNR Bio-Imaging Systems, Ltd., Jerusalem, Israel). Enhanced chemiluminescence (ECL) signals of site-specific phosphorylation of cTnI and cMyBP-C were normalized to corresponding Western immunoblot stains. Samples were typically loaded at least in duplicates (from *n* = 4–6 different rat hearts for each group) during at least 4 independent runs for a given protein assay. All original scans and the number of Western immunoblot observations (Appendix A) are detailed in the Appendix A.

### 2.4. Histology and Immunohistochemistry

Sections of the myocardial tissue samples (5 μm thick, both LV and RV) were deparaffinized and stained for 3-nitrotyrosine (3-NT) or cGMP in immunohistochemistry assays according to previous examinations [18,20].

Immunohistochemistry for 3-NT was performed to assess the presence of nitro-oxidative stress in our model groups. After deparaffinization and blocking, LV and RV sections were incubated in the presence of 3-NT primary antibody (1:80, #10189540, Cayman Chemical, Ann Arbor, MI, USA) at 4 °C overnight, and thereafter secondary antibodies were applied (#MP-7401, ImmPRESS™ HRP peroxidase conjugated-anti-Rabbit IgG, Polymer Detection Kit, Vector Laboratories, Burlingame, CA, USA) for 30 min at room temperature. Black colored nickel-cobalt-enhanced 3,3’-diaminobenzidine (DAB, 6 min, room temperature, Vector Laboratories, Burlingame, CA, USA) was used for the development.

For the determination of myocardial cGMP levels, LV and RV sections were deparaffinized and following antigen retrieval (in citric acid buffer) and blocking, sections were probed with an anti-cGMP antibody (1:200, #ab12416, Abcam, Cambridge, UK). After washing, slides were incubated in the presence of secondary anti-rabbit antibody (Biogenex SuperSensitive Link HK-9R Kit, BioGenex, San Ramon, CA, USA) and developed using Fast Red (Dako, Glostrup, Denmark).

Images of five identical areas of each section (free wall of LV and RV) were taken using a light microscope at 200× (3-NT) or 400× (cGMP) magnifications (Zeiss AxioImager.A1 coupled with Zeiss AxioCAm MRc5 CCD camera, Carl Zeiss, Jena, Germany). 3-NT and cGMP positive areas were determined by color (dark grey and red) thresholding with the ImageJ software (National Institutes of Health, Bethesda, MD, USA) by an independent investigator. The percentages of positively stained tissue areas to total areas (fractional area) were calculated.

### 2.5. Data Analysis and Statistics

Ca^2+^-evoked contractures of myocyte-sized preparations were recorded with a custom-built LABVIEW Data Acquisition platform and analyzed with LabVIEW analyzing software (National Instruments, Austin, TX, USA). Signal intensities of protein bands were quantified by using the ImageJ image processing program (National Institutes of Health, Bethesda, MD, USA) and Magic Plot (Magicplot Systems, Saint Petersburg, Russia) software. Statistical significance was tested by analysis of variance (ANOVA followed by Bonferroni’s post hoc test) by GraphPad Prism 6.0 (GraphPad Software Inc., San Diego, CA, USA) software to evaluate the results. Values are given as mean ± SEM. Statistical significance was accepted at *p* < 0.05.

## 3. Results

### 3.1. Vardenafil Prevented the Reduction of Tissue cGMP Levels and the Increase in 3-NT Generation in ZDF Hearts

cGMP levels and the extent of nitro-oxidative stress were tested by immunohistochemistry. cGMP specific staining intensity was less in both LV and RV tissue samples of ZDF hearts than in those of the Lean or Lean + Vard groups (Figure 1A,B). Conversely, levels of 3-NT were higher in LV and RV tissue samples of ZDF animals than in those of the Lean or Lean + Vard groups (Figure 1C,D). Vardenafil treatment effectively opposed the above alterations in both ventricles of ZDF animals (Figure 1A–D).

### 3.2. PDE-5A Inhibition Prevented the Increase in Passive Tension (F_passive_) in LV and RV Cardiomyocytes in ZDF Animals

F_passive_ was significantly higher in both LV and RV cardiomyocytes of the untreated ZDF group than in those of the control groups. Additionally, the increase in F_passive_ seen in the ZDF group was prevented by facilitating the cGMP-PKG pathway by PDE-5A inhibition, while vardenafil was without effects in both LV and RV cardiomyocytes in the Lean group (Figure 2A,B).

Titin phosphorylation was tested by a phosphoprotein staining assay. The overall phosphorylation level of titin was similar in all groups and for both ventricles, independent of disease or vardenafil treatment (Figure 2C,D).

### 3.3. Ca^2+^-Sensitivity of Force Production Showed Interventricular Differences in ZDF Rats

Ca^2+^-activated force generation (F_active_) was tested in permeabilized LV and RV cardiomyocytes at different Ca^2+^ concentrations (pCa: 4.75–7.0. Maximal Ca^2+^-activated force (F_max_ at pCa 4.75) was similar in all groups for both ventricles, independent of disease or vardenafil treatment (Figure 3A,B). However, Ca^2+^-sensitivity of isometric force production (pCa_50_) of LV cardiomyocytes was significantly higher in ZDF rats than in Lean animals, and this difference was reverted by vardenafil treatment (Figure 3C,E). In contrast, the differences in the means of pCa_50_ values of RV cardiomyocytes did not reach statistical significance among the Lean, Lean + Vard, and untreated ZDF groups. Nevertheless, long-lasting PDE-5A inhibition significantly reduced pCa_50_ of RV cardiomyocytes in the ZDF + Vard group (Figure 3D,F).

### 3.4. Interventricular Differences of Myofilament Protein Phosphorylations

To elucidate the molecular mechanisms leading to altered pCa_50_ values of LV and RV cardiomyocytes in ZDF rats, site-specific phosphorylation studies were performed in myofilament proteins. These assays revealed cTnI hypophosphorylation at Ser-22/23, Ser-43 and Thr-144 phosphosites in LV cardiomyocytes of the untreated ZDF group. Hypophosphorylation of cTnI was prevented at the above cTnI phosphosites by vardenafil treatment in LV cardiomyocytes in the ZDF + Vard group (Figure 4A,C,E). Moreover, vardenafil also increased cTnI phosphorylation in the Lean + Vard group, although the difference between the Lean and Lean + Vard groups reached statistical significance only for the Thr-144 phosphosite (Figure 4E).

RV levels of cTnI phosphorylation at Ser-22/23, Ser-43 and Thr-144 sites were similar in the ZDF and Lean groups (Figure 4B,D,F). Moreover, PDE-5A inhibition did not affect cTnI phosphorylation at Ser-22/23 in the RV of the Lean + Vard and ZDF + Vard groups (Figure 4B). Surprisingly, cTnI hyperphosphorylation was observed at the Ser-43 and Thr-144 sites in the RV of ZDF rats after vardenafil treatment (Figure 4D,F). cTnI phosphorylation was not affected by vardenafil treatment at the Ser-43 and Thr-144 phosphosites in the Lean + Vard group (Figure 4D,F).

cMyBP-C hypophosphorylation was observed at the Ser-282 phosphosite in LV cardiomyocytes of ZDF rats. PDE-5A inhibition prevented cMyBP-C hypophosphorylation at the Ser-282 phosphosite in LV cardiomyocytes in ZDF + Vard group (Figure 5A). No significant differences were observed in cMyBP-C phosphorylation at the Ser-282 site in RV cardiomyocytes in the four experimental groups (Figure 5B).

## 4. Discussion

Results of this study demonstrate uniform increases in F_passive_ and chamber specific alterations in Ca^2+^-sensitivity of force production (pCa_50_) of left and right ventricular cardiomyocytes of ZDF animals. Distinct myofilament protein phosphorylation patterns of Lean and ZDF hearts in the absence and presence of chronic preventive PDE-5A inhibition helped to explain these differences.

The RV is thought to be more vulnerable to hemodynamic and nitro-oxidative stress than the LV during the progression of heart failure [21]. Nevertheless, despite the faster progression of RV to failure than that of LV, RV remodeling is considered to be highly reversible [22]. The above considerations implicate distinct adaptive mechanisms to control RV and LV remodeling when presumably both ventricles are exposed to high levels of hemodynamic and nitro-oxidative stress [9,21].

A characteristic elevation in cardiomyocyte F_passive_ of endomyocardial biopsies of HFpEF patients or ZDF rats provides a plausible link between cardiomyocyte mechanics and LV diastolic dysfunction [18,23]. This stiffening of cardiomyocytes in HFpEF has been repeatedly linked to hypophosphorylation of the giant sarcomeric protein, titin [12,24]. In vitro incubations of permeabilized cardiomyocytes in the presence of protein kinase A (PKA) [25] or PKG [26] increased titin phosphorylation and reduced passive stiffness in human cardiomyocytes. Likewise, stiffening of RV cardiomyocytes could be prevented by enhancing the cGMP-PKG signaling pathway by in vivo vardenafil treatment as demonstrated here and for the LV before [18]. However, the overall phosphorylation level of titin was apparently not affected in the vardenafil treated experimental groups in this study. One reason behind this discrepancy might be that titin possesses a number of phosphosites across its elastic regions [24]. Consequently, multiple protein kinases and phosphatases can act on titin’s elasticity with potential complementary effects on its overall phosphorylation level [27]. Alternatively, oxidative post-translational titin modifications [24] can also explain a constant titin phosphorylation level upon chronic vardenafil administrations. Indeed, results of this study verified that the 3-NT level (a marker of nitro-oxidative stress) can be mitigated by PDE-5A inhibition not only for the LV but also for the RV in ZDF animals [18].

F_max_ was not affected in LV or RV cardiomyocytes of ZDF rats, although T2DM has been formerly associated with RV systolic dysfunction in HFpEF [28,29]. pCa_50_ was increased in LV cardiomyocytes, but it remained unchanged in RV cardiomyocytes in ZDF animals. Nevertheless, unlike in Lean controls, pCa_50_ was effectively decreased in cardiomyocytes of both ventricles by vardenafil treatments in ZDF rats. Similarly, LV and RV cGMP levels were elevated by long-term vardenafil treatment in ZDF animals, but not in Lean controls. These data suggest a more significant role for PDE-5A in the presence of diabetic cardiomyopathy than in its absence, and thus they are consistent with an elevated myocardial expression of PDE-5E in both ventricles during oxidative stress and heart failure [13].

pCa_50_ is known to be regulated by myofilament protein (i.e., cTnI and cMyBP-C) phosphorylation at multiple phosphosites. For example, hypophosphorylation of cTnI at the Ser-22/23 sites results in an increase in pCa_50_ in human systolic HF [30]. Likewise, in an experimental HFpEF model, LV cardiomyocytes of aged hypertensive dogs showed increased pCa_50_, which was associated with hypophosphorylation of cTnI at Ser-22/23. The Ser-22/23 residues of cTnI can be targeted by PKA or PKG [31]. In this study, cTnI hypophosphorylation at Ser-22/23 of LV cardiomyocytes could be prevented by the enhancement of the cGMP-PKG pathway by long-term PDE 5A inhibition (vardenafil therapy) in vivo. Therefore, the PDE-5A inhibition evoked increase in cGMP level (with or without parallel changes in cyclic adenosine monophosphate (cAMP) level) was effective to oppose cTnI hypophosphorylation at Ser-22/23 in LV cardiomyocytes [32]. As a matter of fact, Ser-22/23, Ser-43/45 and Thr-144 phosphosites are also all potential substrates for certain PKC isoforms [33]. In contrast to phosphorylation of Ser-22/23, phosphorylation of Ser-43/45 is expected to decrease F_max_. Former studies demonstrated that PKCα-mediated hyperphosphorylation of cTnI at Ser-43/45 leads to a reduction in F_max_ and pCa_50_ of LV cardiomyocytes from failing hearts of rodents [34,35] and humans [36]. In contrast, cTnI phosphorylation at its Thr-144 site alone might have little effect on F_max_ or pCa_50_ [37]. An interaction between PKC mediated cTnI phosphorylation and vardenafil administration might be attributed to a potential link between cGMP-PKG signaling and cardiomyocyte redox balance. In fact, oxidative stress-dependent substrate preferences have been reported for PKCδ that involved the Ser-22/23 and Thr-144 phosphosites of cTnI [30]. Furthermore, H_2_O_2_ as a mimic of oxidative stress also affected PKC-mediated myofilament phosphorylation and Ca^2+^-sensitivity of force production in the rat [38]. While the involvement of the above mechanisms in our present study is not entirely clear, their end-results in cTnI and cMyBP-C phosphorylations showed clear differences for the LV and RV. In particular, phosphorylation levels of RV cardiomyocytes at cTnI at Ser-43 and at Thr-144 were not changed in the ZDF group, but they were increased beyond control levels following vardenafil treatment, thus contrasting LV cardiomyocytes. 

Phosphorylation of cMyBP-C was shown to be cardioprotective against ischemic injury [39], and phosphorylation of cMyBP-C at Ser-282 might have a unique regulatory role in maintaining normal sarcomere contractility and structure [40]. Ser-282 of cMyBP-C is a substrate for PKA and Ca^2+^/calmodulin-activated kinase II (CaMKII) [40], being downregulated in the failing human heart [41]. Here we report cMyBP-C hypophosphorylation at its Ser-282 site in LV cardiomyocytes of HFpEF rats, which could be prevented by PDE-5A inhibition with vardenafil [42]. Interestingly, phosphorylation of cMyBP-C at Ser-282 remained unchanged in the RV, independent of disease or treatment.

## 5. Conclusions

In previous studies, our group reported cell-to-cell [43], regional [44] and interventricular [45] heterogeneity of phosphorylation and function of regulatory myofilament proteinssuch as cTnI and cMyBP-Cin different HFrEF models. Here we extend the above observations for a rodent model of HFpEF with diabetic cardiomyopathy. Accordingly, while we cannot exclude potential interplays between multiple kinases and redox signaling pathways, our data implicate distinct baseline signaling intensities for the two ventricles in ZDF rats. In general, cardiomyocyte responses to vardenafil treatments were consistent with an effective cGMP-PKG signaling mechanism in diabetic animals for both ventricles. Moreover, our pCa_50_ and myofilament phosphorylation data implicated distinct pharmacological responses and potentially divergent consequences on the systolic functions of the RV and the LV by long-lasting PDE-5A inhibition. Conversely, our data also suggested that boosting the cGMP-PKG signaling pathway might prevent LV and RV diastolic dysfunction in a uniform fashion during diabetic cardiomyopathy and/or HFpEF.

## Figures and Tables

**Figure 1 antioxidants-10-01776-f001:**
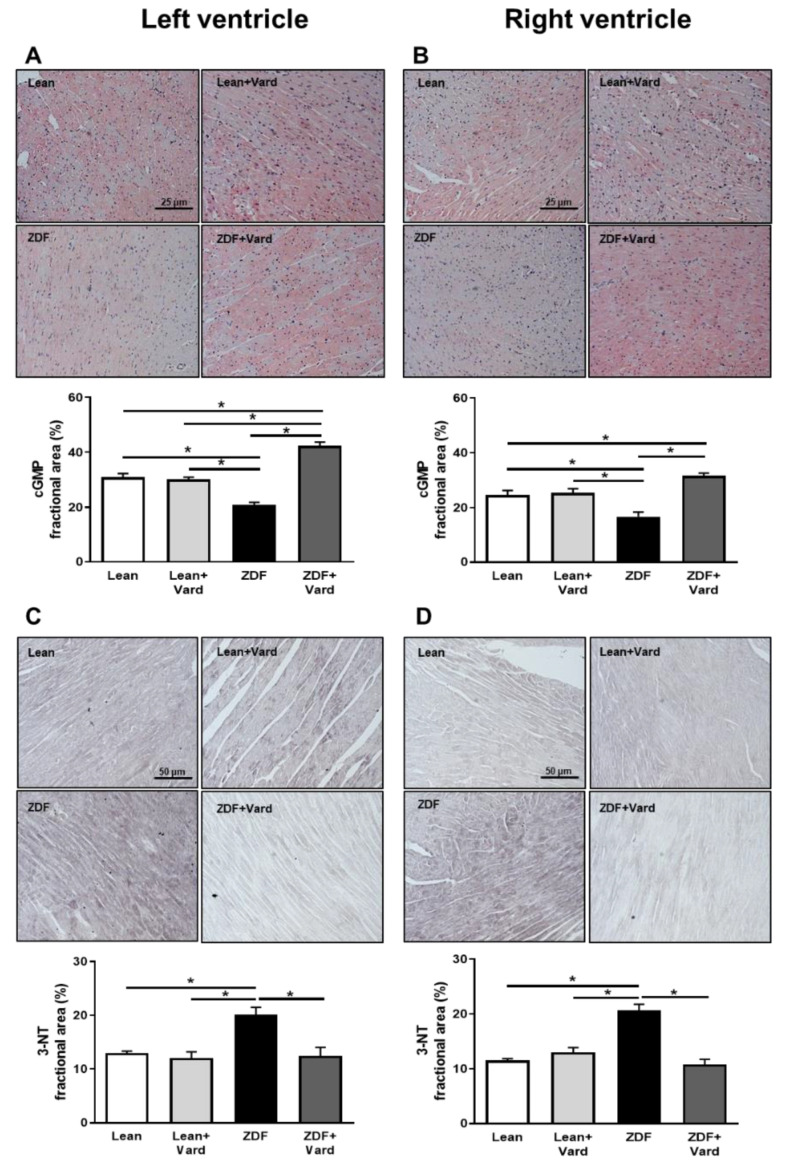
PDE-5A inhibition affected cGMP and 3-NT levels in the left and right ventricles of ZDF hearts. Representative immunohistochemical images of myocardial cGMP ((**A**,**B**), magnification: 400×) and 3-NT ((**C**,**D**), magnification: 200×) generation in LV (**A**,**C**) and RV (**B**,**D**) tissue sections for all experimental groups. Bar graphs show the results of quantitative evaluations for the cGMP assays (^LV^Lean, 31 ± 1.3%; ^LV^Lean + Vard, 30. ± 0.8%; ^LV^ZDF, 20.8 ± 1%; ^LV^ZDF + Vard, 42.3 ± 1.4%; ^RV^Lean, 24.6 ± 1.6%; ^RV^Lean + Vard, 25.3 ± 1.5%; ^RV^ZDF, 16.7 ± 1.7%; ^RV^ZDF + Vard, 31.6 ± 1%; (**A**,**B**) bottom panels) and 3-NT assays (^LV^Lean, 13 ± 0.3%; ^LV^Lean + Vard, 12 ± 1.2%; ^LV^ZDF, 20.1 ± 1.4; ^LV^ZDF + Vard, 12.4 ± 1.6%; ^RV^Lean, 11.5 ± 0.3%; ^RV^Lean + Vard, 12.9 ± 0.9%; ^RV^ZDF, 20.7 ± 1.1%; ^RV^ZDF + Vard, 10.8 ± 0.9%; (**C**,**D**) bottom panels). Values denote fractional areas (%) for all experimental groups. The number (*n*) of rat hearts involved were: Lean, 6–8; Lean + Vard, 4–7; ZDF, 6–8 and ZDF + Vard, 6–8. Data are given as mean ± SEM, * *p* < 0.05. cGMP: cyclic guanosine monophosphate; 3-NT: 3-nitrotyrosine.

**Figure 2 antioxidants-10-01776-f002:**
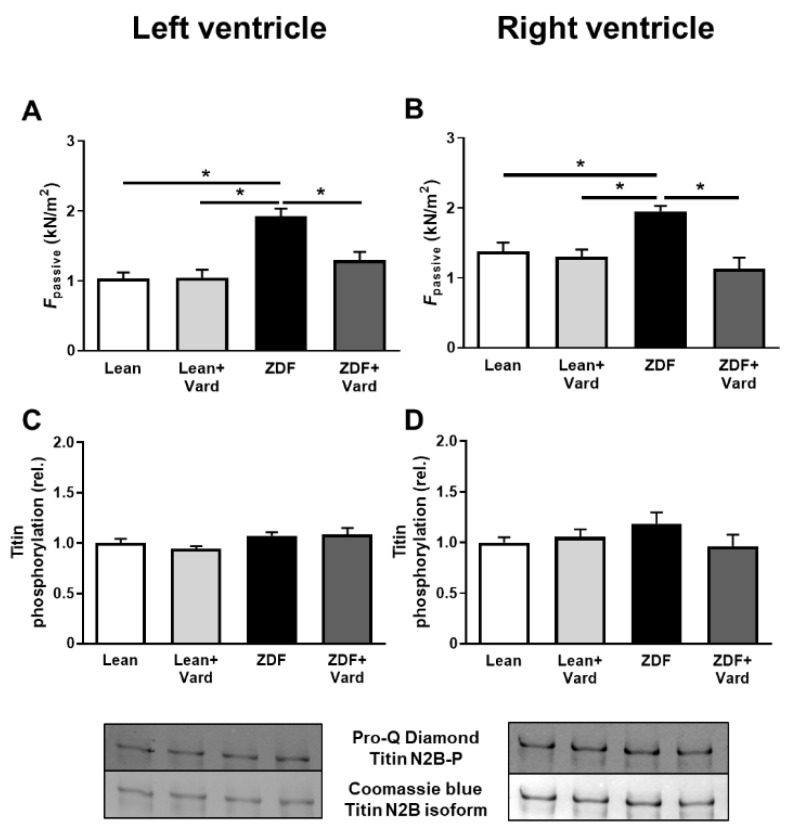
Cardiomyocyte passive tension (F_passive_) and titin protein phosphorylation in the left (**A**,**C**) and right (**B**,**D**) ventricles in the four experimental groups. F_passive_ was measured at a sarcomere length of 2.3 μm (**A**,**B**) (*n* = 6–11 cardiomyocytes from at least three different hearts/groups). Overall phosphorylation level of the N2B titin isoform was assessed by Pro-Q^®^ Diamond phosphoprotein staining. Titin amounts were visualized by Coomassie-blue staining (**C**,**D**). F_passive_ values: ^LV^Lean, 1.03 ± 0.09 kN/m^2^; ^LV^Lean + Vard, 1.05 ± 0.11 kN/m^2^; ^LV^ZDF, 1.98 ±0.12 kN/m^2^; ^LV^ZDF + Vard, 1.30 ± 0.12 kN/m^2^; ^RV^Lean, 1.38 ± 0.12 kN/m^2^; ^RV^Lean + Vard, 1.29 ± 0.11 kN/m^2^; ^RV^ZDF, 1.95 ± 0.08 kN/m^2^, ^RV^ZDF + Vard, 1.13 ± 0.16 kN/m^2^). Overall titin phosphorylation levels: ^LV^Lean, 1.00 ± 0.04; ^LV^Lean + Vard, 0.94 ± 0.03; ^LV^ZDF, 1.07 ± 0.03; ^LV^ZDF + Vard, 1.09 ± 0.06; ^RV^Lean, 1.00 ± 0.05; ^RV^Lean + Vard, 1.05 ± 0.08; ^RV^ZDF, 1.19 ± 1.11; ^RV^ZDF + Vard, 0.96 ± 0.11; all in relative units. Data are given as mean ± SEM, * *p* < 0.05.

**Figure 3 antioxidants-10-01776-f003:**
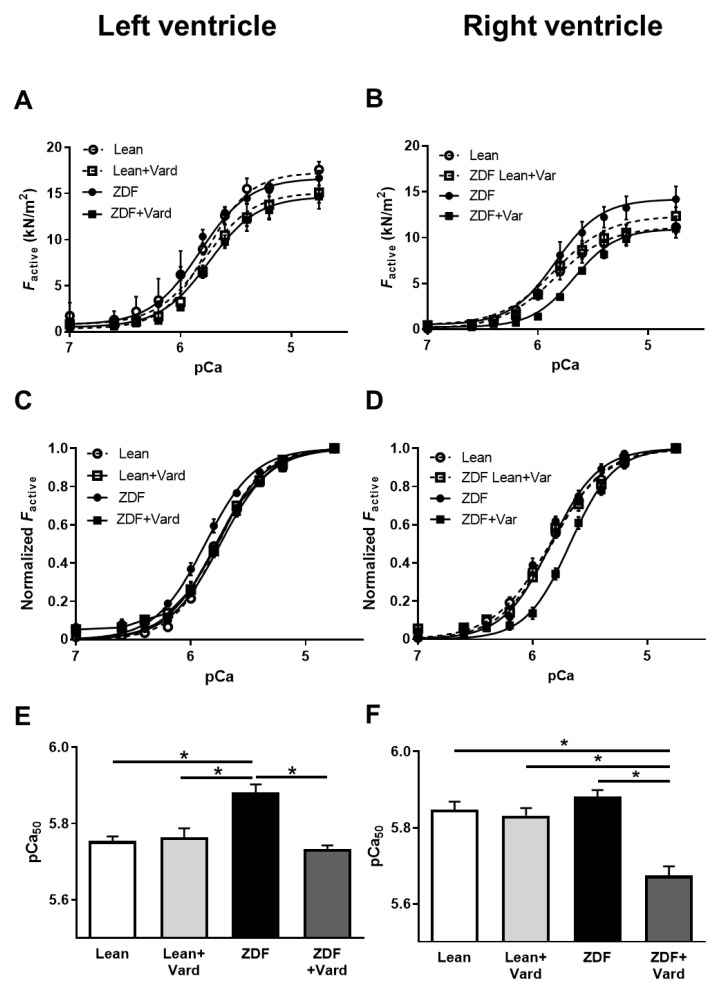
Ca^2+^-sensitivity of isometric force production (pCa_50_) in isolated, permeabilized left ventricular (**A**,**C**,**E**) and right ventricular (**B**,**D**,**F**) cardiomyocytes of rat hearts. Means of active force values (F_active_) are given at different Ca^2+^ concentrations for the left (**A**) and right (**B**) ventricles from the Lean, Lean + Vard, ZDF, and ZDF + Vard groups. Average sarcomere length was set to 2.3 µm. F_active_ values at submaximal Ca^2+^ concentrations (pCa 5.4–7.0) were normalized to F_max_ (pCa 4.75) to illustrate differences in pCa_50_ values (**C**,**D**). Bar graphs (**E**,**F**) show the results of statistical analyses on pCa_50_ values of the four experimental groups. F_max_ (pCa 4.75): ^LV^Lean, 17.59 ± 0.85 kN/m^2^; ^LV^Lean + Vard, 15.11 ± 1.75 kN/m^2^; ^LV^ZDF, 16.67 ± 0.79 kN/m^2^; ^LV^ZDF + Vard, 14.68 ± 0.69 kN/m^2^; ^RV^Lean, 11.15 ± 1.17 kN/m^2^; ^RV^Lean + Vard, 12.36 ± 0.97 kN/m^2^; ^RV^ZDF, 14.20 ± 1.39 kN/m^2^; ^RV^ZDF + Vard, 10.96 ± 0.49 kN/m^2^. pCa_50_: ^LV^Lean, 5.76 ± 0.02; ^LV^Lean + Vard, 5.77 ± 0.02; ^LV^ZDF, 5.88 ± 0.02; ^LV^ZDF + Vard, 5.73 ± 0.01; ^RV^Lean, 5.85 ± 0.02; ^RV^Lean + Vard, 5.83 ± 0.02; ^RV^ZDF, 5.88 ± 0.01; ^RV^ZDF + Vard, 5.67 ± 0.03. pCa = −lg[Ca^2+^] where [Ca^2+^] is the molar concentration of Ca^2+^. Data are expressed as mean ± SEM, * *p* < 0.05; *n* = 6–11 cardiomyocytes from at least three different hearts.

**Figure 4 antioxidants-10-01776-f004:**
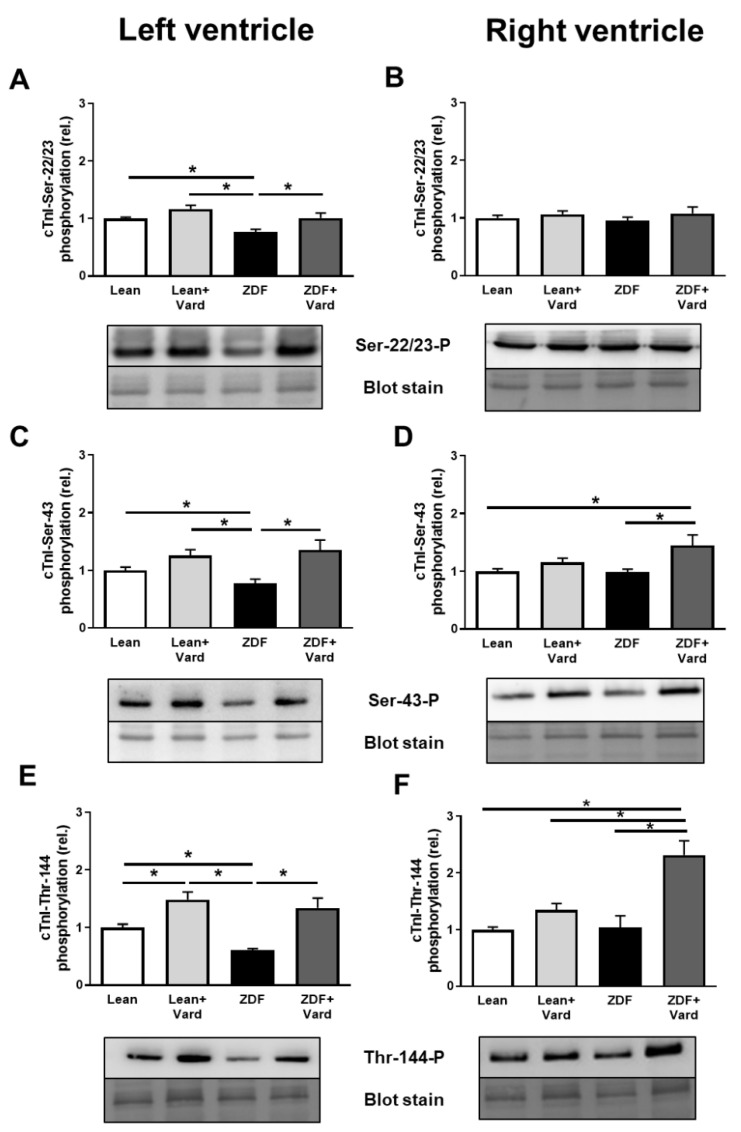
Results of site-specific phosphorylation assays for cTnI in the left (**A**,**C**,**E**) and right ventricles (**B**,**D**,**F**). Cardiac troponin-I (cTnI) phosphorylation levels at the Ser-22/23 (**A**,**B**), Ser-43 (**C**,**D**) and Thr-144 amino acid residues (**E**,**F**) were tested by Western immunoblotting in the Lean, Lean + Vard, ZDF, and ZDF + Vard groups for both ventricles. cTnI phosphorylation levels at Ser-22/23: ^LV^Lean, 1.00 ± 0.02; ^LV^Lean + Vard, 1.17 ± 0.06; ^LV^ZDF, 0.77 ± 0.04; ^LV^ZDF + Vard, 1.01 ± 0.09; ^RV^Lean, 1.00 ± 0.05; ^RV^Lean + Vard, 1.07 ± 0.06; ^RV^ZDF, 0.95 ± 0.07; ^RV^ZDF + Vard, 1.08 ± 0.11; at Ser-43: ^LV^Lean, 1.00 ± 0.06; ^LV^Lean + Vard, 1.26 ± 0.10; ^LV^ZDF, 0.77 ± 0.08; ^LV^ZDF + Vard, 1.35 ± 0.17; ^RV^Lean, 1.00 ± 0.04; ^RV^Lean + Vard, 1.16 ± 0.07; ^RV^ZDF, 0.98 ± 0.05; ^RV^ZDF + Vard, 1.45 ± 0.19; at Thr-144: ^LV^Lean, 1.00 ± 0.06; ^LV^Lean + Vard, 1.48 ± 0.13; ^LV^ZDF, 0.61 ± 0.03; ^LV^ZDF + Vard, 1.34 ± 0.17; ^RV^Lean, 1.00 ± 0.05; ^RV^Lean + Vard, 1.35 ± 0.12; ^RV^ZDF, 1.04 ± 0.21; ^RV^ZDF + Vard, 2.32 ± 0.25, all in relative units. Phosphorylation sites of cTnI were labelled with specific antibodies. Total protein amounts were assessed by a super sensitive blot stain. Bars represent means ± SEM (normalized to the mean of the Lean group as control), * *p* < 0.05, *n* = 9–15 independent determinations from at least four different hearts.

**Figure 5 antioxidants-10-01776-f005:**
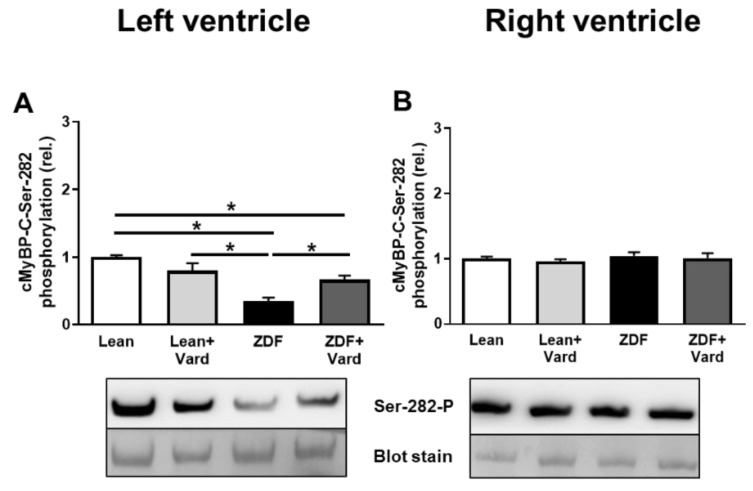
Phosphorylation level of cMyBP-C at the Ser-282 phosphosite in cardiomyocytes of left ventricles (**A**) and right ventricles (**B**) of the experimental groups. Phosphorylation levels of cMyBP-C were assayed by Pro-Q^®^ Diamond phosphoprotein staining. The upper bands illustrate the phosphorylation status of cMyBP-C and the lower bands indicate the total protein amounts for representative membranes. cMyBP-C phosphorylation levels at the Ser-282 phosphosite: ^LV^Lean, 1.00 ± 0.03; ^LV^Lean + Vard, 0.80 ± 0.11; ^LV^ZDF, 0.35 ± 0.06; ^LV^ZDF + Vard, 0.67 ± 0.06; ^RV^Lean, 1.00 ± 0.03; ^RV^Lean + Vard, 0.96 ± 0.03; ^RV^ZDF, 1.05 ± 0.06; ^RV^ZDF + Vard, 1.01 ± 0.08, all in relative units. *n* = 8–10 independent determinations from at least four different heart samples. Values are given as mean ± SEM, * *p* < 0.05 vs. control (Lean) group.

## Data Availability

Data is contained within the article and Appendix A.

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
