# Peer review of "Long-Term PDE-5A Inhibition Improves Myofilament Function in Left and Right Ventricular Cardiomyocytes through Partially Different Mechanisms in Diabetic Rat Hearts"

_antioxidants, 2021, doi:10.3390/antiox10111776_

Round 1
Reviewer 1 Report
The manuscript of “Inhibition of cGMP breakdown improves myofilament function in left and right ventricular cardiomyocytes through different mechanisms of diabetic rats” by B. Bódi et al. aims to characterize the right and left ventricular remodeling and their prevention by vardenafil (a long-acting phos-21 phodiesterase-5A (PDE-5A) inhibitor) administration in a diabetic cardiomyopathy model. The authors demonstrated the distinct roles of PDE-5A regulated cyclic guanosine monophosphate 37 (cGMP)/protein kinase G signaling in the left and right ventricles of diabetic rats, which may be involved in the progression of heart failure with preserved ejection fraction in diabetic cardiomyopathy. The manuscript may be accepted for publication after major revision.
Comments:
- The authors should provide original scans of all WBs in the Supplementary material section. Western blots report data were obtained from a too small cohort (n = 3). As Western blots can have substantial variations, depending on load and blot transfer, I urge the authors to use more animals for this analysis.
- The Abstract section and the Results section are overloaded with numerical values (means and errors) for each group, which makes it difficult to understand the text. So, please remove numerical values where it is possible.
- In the abstract section, it is necessary to emphasize the differences in mechanisms involved in the improvement of myofilament function in left and right ventricular cardiomyocytes in long term PDE-5A inhibition (vardenafil therapy) in vivo. In the Discussion section, the authors should discuss in more detail the possible applications of the differences.
- To improve the Discussion, the indexes of oxidative stress in the groups studied need to be measured.
- The English is not good enough for publication. The manuscript should be carefully revised by native English speakers/a professional language editing service to improve the readability. The title of the manuscript should be improved.
Author Response
Reviewer #1:
The manuscript of “Inhibition of cGMP breakdown improves myofilament function in left and right ventricular cardiomyocytes through different mechanisms of diabetic rats” by B. Bódi et al. aims to characterize the right and left ventricular remodeling and their prevention by vardenafil (a long-acting phos-21 phodiesterase-5A (PDE-5A) inhibitor) administration in a diabetic cardiomyopathy model. The authors demonstrated the distinct roles of PDE-5A regulated cyclic guanosine monophosphate 37 (cGMP)/protein kinase G signaling in the left and right ventricles of diabetic rats, which may be involved in the progression of heart failure with preserved ejection fraction in diabetic cardiomyopathy. The manuscript may be accepted for publication after major revision.
Response: Thank you for considering our work potentially acceptable after major revision. We are also grateful for your critical remarks because they largely improved our manuscript.
1/ The authors should provide original scans of all WBs in the Supplementary material section. Western blots report data were obtained from a too small cohort (n = 3). As Western blots can have substantial variations, depending on load and blot transfer, I urge the authors to use more animals for this analysis.
Response: We are sorry for giving a wrong impression on the size of the cohort for Western immunoblot analyses in the previous version of the manuscript. We strongly agree that Western immunoblots can have substantial variations, depending on load and blot transfer. To minimize potential errors due to these factors, samples were loaded typically in duplicates or triplicates from n=4-6 different hearts. The numbers of animals and repeated assays have been clarified for each protein analyses in a table in the new Supplementary material section (Table S1). Moreover, now we also show original scans of all Western blots in the Supplementary material section as requested by this reviewer.
Change in text (Materials and Methods, SDS-PAGE and Western immunoblot analysis, Supplementary material): “Samples were typically loaded in duplicates or triplicates (from n = 4-6 different rat hearts for each group) during at least 4 independent runs for a given protein assay. All original scans and the number of Western immunoblot observations (Table S1) are detailed in the Supplementary material.” (Materials and Methods)
A new Supplementary material file with all original scans and a table detailing the number of repeated assays for each experimental group has been included.
2/ The Abstract section and the Results section are overloaded with numerical values (means and errors) for each group, which makes it difficult to understand the text. So, please remove numerical values where it is possible.
Response: We agree and accepted your criticism. In line with your request, numerical values have been removed from the Abstract/Result sections, now they appear in the legends (and now they follow a uniform structure).
Change in text (Abstract, Results, Figure Legends): “Vardenafil treatment increased cyclic guanosine monophosphate (cGMP) levels and decreased 3-nitrotyrosine (3-NT) levels in the left and right ventricles of ZDF animals, but not in Lean animals. Cardiomyocyte passive tension (Fpassive) was higher in LV and RV cardiomyocytes of ZDF rats than in those receiving preventive vardenafil treatment. Levels of overall titin phosphorylation did not differ in the four experimental groups. Maximal Ca2+-activated force (Fmax) of LV and RV cardiomyocytes were preserved in ZDF animals. Ca2+-sensitivity of isometric force production (pCa50) was significantly higher in LV (but not in RV) cardiomyocytes of ZDF rats than in their counterparts in the Lean or Lean+Vard groups. In accordance, the phosphorylation levels of cardiac troponin I (cTnI) and myosin binding protein-C (cMyBP-C) were lower in LV (but not in RV) cardiomyocytes of ZDF animals than in their counterparts of the Lean or Lean+Vard groups. Vardenafil treatment normalized pCa50 values in LV cardiomyocytes, and it decreased pCa50 below control levels in RV cardiomyocytes in the ZDF+Vard group.” (Abstract)
Numerical values were removed from the Results and now they appear in Figure Legends.
3/ In the abstract section, it is necessary to emphasize the differences in mechanisms involved in the improvement of myofilament function in left and right ventricular cardiomyocytes in long term PDE-5A inhibition (vardenafil therapy) in vivo. In the Discussion section, the authors should discuss in more detail the possible applications of the differences.
Response: We are very grateful for this comment, because it encouraged us to elaborate the implications of our findings. Accordingly, now we stress more explicitly the similarities and also the differences between LV and RV cardiomyocytes of ZDF animals and ZDF+Vard animals, and now we highlight their hypothetical in vivo implications (Abstract and Discussion).
Change in text (Abstract, Discussion): “Our data illustrate partially overlapping myofilament protein alterations for LV and RV cardiomyocytes in diabetic rat hearts upon long-term PDE-5A inhibition. While uniform patterns in cGMP, 3-NT and Fpassive levels predict identical effects of vardenafil therapy for the diastolic function in both ventricles, the uneven cTnI, cMyBP-C phosphorylation levels and pCa50 values implicate different responses for the systolic function.” (Abstract)
“Moreover, our pCa50 and myofilament phosphorylation data implicated distinct pharmacological responses and potentially divergent consequences on the systolic functions of the RV and the LV by long-lasting PDE-5A inhibition. Conversely, our data also suggested that boosting the cGMP-PKG signaling pathway might prevent LV and RV diastolic dysfunction in a uniform fashion during diabetic cardiomyopathy and/or HFpEF.” (Discussion)
4/ To improve the Discussion, the indexes of oxidative stress in the groups studied need to be measured.
Response: We accepted this criticism and included new experimental data illustrating oxidative stress. To this end, an immunohistochemical study was performed in all experimental groups where the tissue levels of 3-nitrotyrosine (3-NT) were screened. Results of these measurements revealed increased 3-NT levels in both LV and RV chambers of ZDF hearts, and that this effect could be prevented by long-lasting vardenafil treatment.
Change in text (Abstract, Material and Methods, Results, Discussion, Figures, Legends): “In vitro force measurements, biochemical and histochemical assays were employed to assess cardiomyocyte function and signaling. Vardenafil treatment increased cyclic guanosine monophosphate (cGMP) levels and decreased 3-nitrotyrosine (3-NT) levels in the left and right ventricles of ZDF, but not in Lean animals.” (Abstract)
“Sections of the myocardial tissue samples (5 μm thick, both LV and RV) were deparaffinized and stained for 3-nitrotyrosine (3-NT) or cGMP in immunohistochemistry assays according to previous examinations [18,20].
Immunohistochemistry for 3-NT was performed to assess the presence of nitro-oxidative stress in our model groups. After deparaffinization and blocking, LV and RV sections were incubated in the presence of 3-NT primary antibody (1:80, #10189540, Cayman Chemical, Ann Arbor, MI, USA) at 4 °C overnight, and thereafter secondary antibodies were applied (#MP-7401, ImmPRESS™ HRP peroxidase conjugated-anti-Rabbit IgG, Polymer Detection Kit, Vector Laboratories, Burlingame, CA, USA) for 30 min at room temperature. Black coloured nickel-cobalt-enhanced 3,3'-diaminobenzidine (DAB, 6 min, room temperature, Vector Laboratories, Burlingame, CA, USA) was used for the development.” (Materials and Methods)
“Vardenafil prevented the reduction of tissue cGMP levels and the increase in 3-NT generation in ZDF hearts
cGMP levels and the extent of nitro-oxidative stress were tested by immunohistochemistry. cGMP specific staining intensity was less in both LV and RV tissue samples of ZDF hearts than in those of the Lean or Lean+Vard groups (Figure 1A and 1B). Conversely, levels of 3-NT were higher in LV and RV tissue samples of ZDF animals than in those of the Lean or Lean+Vard groups (Figure 1C and 1D). Vardenafil treatment effectively opposed the above alterations in both ventricles of ZDF animals (Figure 1A-1D).” (Results)
“Alternatively, oxidative post-translational titin modifications [24] can also explain a constant titin phosphorylation level upon chronic vardenafil administrations. Indeed, results of this study verified that 3-NT level (a marker of nitro-oxidative stress) can be mitigated by PDE-5A inhibition not only for the LV but also for the RV in ZDF animals [18].” Discussion
New Figure 1 and new figure legend 1
New reference: [20] Oláh, A.; Németh, B.T.; Mátyás, C.; Horváth, E.M.; Hidi, L.; Birtalan, E.; Kellermayer, D.; Ruppert, M.; Merkely, G.; Szabó, G.; et al. Cardiac effects of acute exhaustive exercise in a rat model. Int J Cardiol 2015, 182, 258-266, doi:10.1016/j.ijcard.2014.12.045.
5/ The English is not good enough for publication. The manuscript should be carefully revised by native English speakers/a professional language editing service to improve the readability. The title of the manuscript should be improved.
Response: We made an effort to improve the English of the text. The title of the paper has been changed.
Change in text (the entire paper, title): “Long-term PDE-5A inhibition improves myofilament function in left and right ventricular cardiomyocytes through partially different mechanisms in diabetic rat hearts” (Title)

Reviewer 2 Report
The study is limited in scope but the work is well done and convincing. The only issue that I have is with the title. While I understand that the connection between Vardenfil and cGMP breakdown has previously been established by the authors and others, this paper contains no direct evidence supporting their conclusion that the affects observed are due inhibition of cGMP breakdown.
Author Response
Reviewer #2:
The study is limited in scope but the work is well done and convincing. The only issue that I have is with the title. While I understand that the connection between Vardenfil and cGMP breakdown has previously been established by the authors and others, this paper contains no direct evidence supporting their conclusion that the affects observed are due inhibition of cGMP breakdown.
Response: Thank you for your positive evaluation. Motivated by your comment now we included cGMP determinations for all experimental groups and also changed the title of the paper.
Change in text (Abstract, Material and Methods, Results, Discussion, Figures, Legends): “In vitro force measurements, biochemical and histochemical assays were employed to assess cardiomyocyte function and signaling. Vardenafil treatment increased cyclic guanosine monophosphate (cGMP) levels and decreased 3-nitrotyrosine (3-NT) levels in the left and right ventricles of ZDF animals, but not in Lean animals.” (Abstract)
“For the determination of myocardial cGMP levels LV and RV sections were deparaffinized and following antigen retrieval (in citric acid buffer) and blocking, sections were probed with an anti-cGMP antibody (1:200, #ab12416, Abcam, Cambridge, UK). After washing, slides were incubated in the presence of secondary anti-rabbit antibody (Biogenex SuperSensitive Link HK-9R Kit, BioGenex, San Ramon, CA, USA) and developed using Fast Red (Dako, Glostrup, Denmark).
Images of five identical areas of each section (free wall of LV and RV) were taken using light microscope at 200x (3-NT) or 400x (cGMP) magnifications (Zeiss AxioImager.A1 coupled with Zeiss AxioCAm MRc5 CCD camera, Carl Zeiss). 3-NT and cGMP positive areas were determined by colour (dark grey and red) thresholding with the ImageJ software (National Institutes of Health, Bethesda, MD, USA) by an independent investigator. The percentages of positively stained tissue areas to total areas (fractional area) were calculated.” (Materials and Methods)
“Vardenafil prevented the reduction of tissue cGMP levels and the increase in 3-NT generation in ZDF hearts
cGMP levels and the extent of nitro-oxidative stress were tested by immunohistochemistry. cGMP specific staining intensity was less in both LV and RV tissue samples of ZDF hearts than in those of the Lean or Lean+Vard groups (Figure 1A and 1B). Conversely, levels of 3-NT were higher in LV and RV tissue samples of ZDF animals than in those of the Lean or Lean+Vard groups (Figure 1C and 1D). Vardenafil treatment effectively opposed the above alterations in both ventricles of ZDF animals (Figure 1A-1D).” (Results)
“Similarly, LV and RV cGMP levels were elevated by long-term vardenafil treatment in ZDF animals, but not in Lean controls. These data suggest a more significant role for PDE-5A in the presence of diabetic cardiomyopathy than in its absence, and thus is consistent with an elevated myocardial expression of PDE-5E in both ventricles during oxidative stress and heart failure [13].” (Discussion)
“Therefore, the PDE-5A inhibition evoked increase in cGMP level (with or without parallel changes in cyclic adenosine monophosphate (cAMP) level) was effective to oppose cTnI hypophosphorylation at Ser-22/23 in LV cardiomyocytes [32].” (Discussion)
New Figure 1 and new figure legend 1
“Long-term PDE-5A inhibition improves myofilament function in left and right ventricular cardiomyocytes through partially different mechanisms in diabetic rat hearts” (Title)

Round 2
Reviewer 1 Report
The authors have significantly improved the manuscript and addressed all my concerns. This manuscript may be accepted for publication now.